# Protocol for an interdisciplinary cross-sectional study investigating the social, biological and community-level drivers of antimicrobial resistance (AMR): Holistic Approach to Unravel Antibacterial Resistance in East Africa (HATUA)

Benon B Asiimwe,[1] John Kiiru,[2] Stephen E Mshana,[3] Stella Neema,[4] Katherine Keenan [ID],[5] Mike Kesby,[5] Joseph R Mwanga,[6] Derek J Sloan,[7] Blandina T Mmbaga,[8] V Anne Smith,[9] Stephen H Gillespie,[7] Andy G Lynch,[7,10] Alison Sandeman,[7] John Stelling,[11] Alison Elliott [ID],[12,13] David M Aanensen,[14,15] Gibson E Kibiki,[16] Wilber Sabiiti,[7] Matthew T G Holden,[7] HATUA Consortium

► Prepublication history and supplemental material for this paper is available online. To view these files, please visit the journal online (http://dx.doi.org/10.1136/bmjopen-2020-041418).

BBA, JK and SEM contributed equally.

For numbered affiliations see end of article.

**Correspondence to**
Dr Katherine Keenan;
katherine.keenan@st-andrews.ac.uk

## ABSTRACT

**Introduction** Antimicrobial resistance (AMR) is a global health threat that requires urgent research using a multidisciplinary approach. The biological drivers of AMR are well understood, but factors related to treatment seeking and the social contexts of antibiotic (AB) use behaviours are less understood. Here we describe the Holistic Approach to Unravel Antibacterial Resistance in East Africa, a multicentre consortium that investigates the diverse drivers of drug resistance in urinary tract infections (UTIs) in East Africa.

**Methods and analysis** This study will take place in Uganda, Kenya and Tanzania. We will conduct geospatial mapping of AB sellers, and conduct mystery client studies and in-depth interviews (IDIs) with drug sellers to investigate AB provision practices. In parallel, we will conduct IDIs with doctors, alongside community focus groups. Clinically diagnosed patients with UTI will be recruited from healthcare centres, provide urine samples and complete a questionnaire capturing retrospective treatment pathways, sociodemographic characteristics, attitudes and knowledge. Bacterial isolates from urine and stool samples will be subject to culture and antibiotic sensitivity testing. Genomic DNA from bacterial isolates will be extracted with a subset being sequenced. A follow-up household interview will be conducted with 1800 UTI-positive patients, where further environmental samples will be collected. A subsample of patients will be interviewed using qualitative tools. Questionnaire data, microbiological analysis and qualitative data will be linked at the individual level. Quantitative data will be analysed using statistical modelling, including Bayesian network analysis, and all forms of qualitative data analysed through iterative thematic content analysis.

### Strengths and limitations of this study

► Multisite, multicountry study with harmonised tools providing opportunity for contextual comparisons.
► Provides novel linkage at the patient level between treatment pathways, sociodemographics, knowledge and attitudes, and pathogen characteristics (antibiotic (AB) susceptibility and genetic profile).
► Describes the AB provision landscape context of the patients through using geospatial mapping, mystery client and qualitative interviews.
► Patient sample is representative of a population of urinary tract infection clinic attendees rather than the general population.

**Ethics and dissemination** Approvals have been obtained from all national and local ethical review bodies in East Africa and the UK. Results will be disseminated in communities, with local and global policy stakeholders, and in academic circles. They will have great potential to inform policy, improve clinical practice and build regional pathogen surveillance capacity.

## INTRODUCTION

Antimicrobial resistance (AMR) emerges when pathogens evolve ways to survive treatments (ie, antibiotic (AB), antiprotozoal, antiviral and antifungal medicines). Antibiotic resistance (ABR) is a significant subset of this phenomenon and is the focus of this study. Increasing levels of resistance to ABs (ABs in this study is specifically used to refer to drugs with antibacterial activity) are a

serious threat to global health and, if no action is taken, are projected to cause 10 million excess deaths by 2050.[1] It is unlikely that this burden will be evenly distributed, and Africa is particularly vulnerable to the challenges posed by AMR/ABR since the continent suffers the highest morbidity and mortality arising from infectious diseases and the least developed laboratory infrastructure.[2] The economic, cultural and ethnic diversity of Africa means that the problems surrounding drug resistance are likely both to be distinct from other regions of the world and to display significant intracontinental diversity. Regional solutions and local approaches are necessary.

Beyond the microbiological and biological origins of ABR and AMR, there are biosocial problems requiring an interdisciplinary approach that incorporate social science perspectives on human–microbial interactions.[3] Despite this, social science perspectives on the evolution and control of AMR are rare.[4] While the biological drivers of AMR in pathogens are well explored,[5] the extent to which these are modulated by human behaviour in and around ABs is less well understood. Cultural, social, economic and clinical factors play a part in shaping the way people source, consume, use and distribute ABs.[6] More effective AB stewardship is acknowledged as one of the key interventions to preserve existing ABs. If this is to be achieved, then we must address the knowledge gap of structural factors and social behaviours that drive pathogens' AB selection pressure and, ultimately, genetic changes in pathogens. Such a problem cannot be achieved by one scientific discipline acting alone. Rather, this complex, multifaceted problem requires an integrative approach able to work effectively across disciplines.

Here, we describe a newly formed consortium—the Holistic Approach to Unravel Antibacterial Resistance (HATUA). The consortium brings together expertise in microbiology, pathogen genomics, epidemiology, human geography, anthropology, sociology, computational biology and statistics across seven institutions, from three East African countries (EACs), the UK and the USA. Taking *hatua* (a Kiswahili word for 'step' or 'action') as inspiration for its acronym, the consortium addresses the social and biological drivers of AB drug resistance in multiple sites in Kenya, Tanzania and Uganda, using the clinical prism of urinary tract infection (UTI). UTIs are common globally, and, in low-income and middle-income countries rarely have a laboratory diagnosis. Moreover, they are often mistaken for other illnesses such as sexually transmitted infections (STIs), and consequently remain poorly treated.[7 8] Through synthesis across multiple sites of our study of the burden and drivers of ABR at national and regional levels, we will provide insights on ABR emergence that may be applicable to other diseases and contexts.

The research will focus on four key elements of the ABR problem in UTI: the therapy landscape, the pathogen, the patient and the community. The research will target three main (inter-related) drivers: first, the supply of ABs; second, the level of knowledge of proper use of ABs; and third, the choice of AB by clinicians and patients and relative effectiveness of treatment. In each case, we will bring novel methodological and theoretical approaches to bear on the issues. Our work will deliver a unique research dataset that links patients, the pathogen and the socio-economic and sociodemographic picture of the individual and their community. Through our research and impact activities, we will also strengthen diagnostic and analytical capacity, and dynamic pathogen surveillance in the region.

## Theoretical and conceptual framework

A challenge of interdisciplinary working is not necessarily finding commonly understood methodologies but shared theoretical (ontological and epistemological) frameworks. This is a theme with which the 'one health' paradigm must grapple if it wishes to understand how infectious disease processes are products of both biological and social relations.[9] In this study, we conceptualise the drivers of AB-resistant UTI infections as part of a complex system of inter-relating biological and social entities,[10] drawing theoretical inspiration from assemblage theory, which facilitates the incorporation of a range of different material actors/actants (humans, animals and microbes) in a single dynamic system.[11] Its advantage is that it encourages experimentalists to engage with the conscious human decision-making in biosocial systems. This approach encourages social scientists to take an 'ontological turn', recognising that 'inanimate' material things (be they bacteria, drugs or clinics) are not merely inert unless given meaning by human subjects, but are themselves able to animate and produce effects.[12] Consequently, we argue that 'new materialist' approaches[13] provide an ideal framework for conceptualising ABR as a complex assemblage of human and non-human entities operating at various scales.

In developing our study, we reviewed the extant literature on possible social and biological drivers and constructed a representation of a UTI-related ABR assemblage (see simplified version figure 1) . The representation maintains the false poles of 'social' and 'biological' but only to emphasise the importance of exploring the integration of sociobiological factors that facilitate ABR. We know that AB suppress bacteria exerting selection pressure that results in the development of ABR (figure 1, blue/right), but some of the drivers of this process lie in the social realm (figure 1, white/left). The behaviours of individual agents, such as healthcare workers, patients and those in the community, are situated within the structures and institutions that shape them. For example, improvements in structural inequalities around water, sanitation and hygiene at the household or community scale might reduce the development of UTIs and the need for antimicrobial use in the first place,[14] removing one significant driver of ABR selective pressure.

Previous research has focused on challenges faced by clinicians in attempting to prescribe antimicrobials effectively, such as knowledge, diagnosis and drug availability.[15]

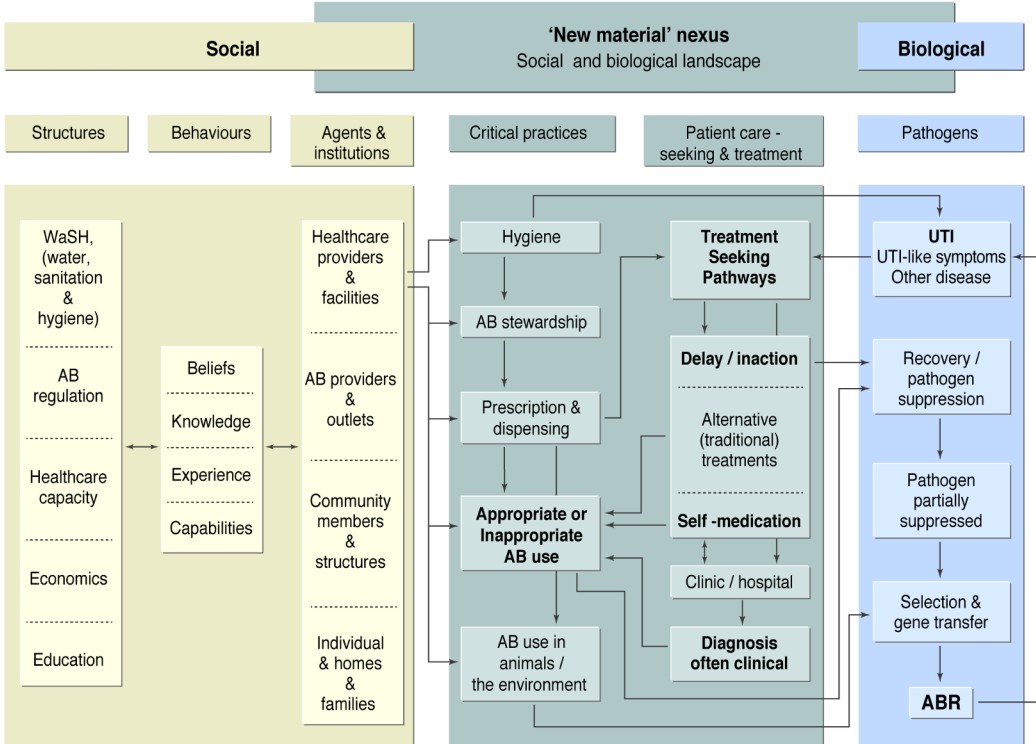

**Figure 1** ABR assemblage (a complex set of inter-related factors) as it refers to UTI. AB, antibiotic; ABR, antibiotic resistance; UTI, urinary tract infection.

However, in EAC, pharmacies, drug shops and other informal sellers are ubiquitous, existing side by side with the public and private healthcare systems, other providers such as traditional healers and veterinary providers. With such medical pluralism, non-prescription dispensing of antimicrobials for self-medication is extremely common,[16 17] influenced not only by individual-level abilities but also by larger social and regulatory structures.

At the centre of this assemblage (figure 1), we identify a 'new material' nexus (coloured grey) consisting of critical practices (such as dispensing, AB use in animals and AB stewardship practices) and the treatment-seeking pathways of patients with UTI. This central nexus describes the entanglement of sociobiological processes that likely drive selective pressure in bacteria and thence development of ABR UTI. A focus for HATUA is to describe and investigate the social and biological drivers of patients' treatment-seeking pathways, or 'patient pathways', and how these relate to patterns of ABR at individual and community levels. We conceptualise a patient pathway as a longitudinal sequence of health-seeking behaviours taken by individuals when they feel ill, which might include delays in seeking treatment, self-medication,[18] attending various formal and informal healthcare providers, and taking medications more or less appropriately. These pathways may be complex, non-linear and, given the economic and sociocultural barriers to clinically ideal pathways, iterative. Rather than theorise healthcare decision making as patients freely and rationally choosing from a suite of available options, pathway-based

models view behaviour as a sequence of steps each with its own situated rationality, and set of social dependencies, constraints and inter-relationships.[19 20] This draws on concepts of 'medical syncretism', which describe how patients may oscillate between different types of healthcare in a single illness episode.[21] Requiring detailed longitudinal analysis, this approach has most often been operationalised using qualitative interviews, as in the case of abortion-related care,[22–24] and quantitative approaches are rare.[25–27] In this study, we collect large-scale quantitative data alongside qualitative in-depth interviews (IDIs), which are linked to individual-level pathogen and ABR profiles.

### Pilot phase 2017–2018
HATUA pilot activities in 2017–2018 in Uganda and Kenya aimed to develop capacity and to demonstrate the feasibility of the holistic study design by conducting a study of microbiological and genomic features of urinary pathogens collected from clinic patients, combined with quantitative sociodemographic data. Patients with UTI-like symptoms were recruited in public clinics and hospitals in Nairobi, Kenya and Isingiro District, Uganda, and provided bacterial samples (most commonly urine, but also stool). In Uganda, 129 of these patients (or their guardians) also completed questionnaires capturing their sociodemographic features, AB knowledge and behavioural characteristics (see summary in online supplemental appendix A, table A1). While the sample size for the questionnaire was not sufficient for detailed

statistical analysis, the data indicated a higher proportion of women recruited than men (71% female), and the majority of patients were of working age. Over half of the respondents had taken medication in the last 6 months (55%), and of those who did, most obtained medications from clinics or health centres and nearly one-fifth (18%) from drug shops.

Urine samples were analysed using culture and antibiotic sensitivity testing (C&AST), and a total of 150 bacterial isolates (n=91 from Kenya and n=59 from Uganda) were genomically characterised, which confirmed a high prevalence of uropathogenic strains *Escherichia coli* and *Klebsiella pneumoniae*, and revealed high levels of multidrug resistance mainly disseminated via clonal and horizontal transfers (the full results are reported here[28]). This exploratory pilot study illustrated the feasibility of collecting linked microbiological, genomic and socioeconomic data, and highlighted important operational fieldwork issues regarding linkage and follow-up to the homestead.

We also conducted IDIs with healthcare providers (eg, nursing assistants) and four focus group discussions (FGDs) with community members (mainly crop farmers and pastoralists) in Isingiro District, Uganda, to explore behaviours and attitudes to AB use and to identify possible drivers of ABR for hypothesis generation. These data were analysed using thematic content analysis, which highlighted potential drivers of ABR. Among community members, these included distrust and misuse of ABs, failure to complete treatment courses, human use of veterinary drugs and combined consumption of ABs and traditional medicines, which informed the development of the research questions in the main study.

## METHODS AND ANALYSIS: MAIN HATUA STUDY
### Field activities and data collection
HATUA's activity will take place in Kenya, Uganda and Tanzania, and multidisciplinary teams will sequentially survey in three study areas (SAs) in each country (figure 2). The locations are sociodemographically distinct: (1) urban, economically advanced settings that potentially increase affordability and access to ABs; (2) remote villages in poorer areas where poverty, physical isolation and lower access to ABs possibly lead to potential drivers, such as sharing of prescriptions, restricted microbiological culture and poor AB sensitivity testing (C&AST) capacity; and (3) pastoralist and neglected network-areas with highly mobile pastoralist communities, and high levels of animal–human interaction fostering a zoonotic link. The tertiary hospital (level 5), and lower-level healthcare facilities in the SAs (level 4, 3 and 2 clinics/hospitals) will be used to recruit patients with UTI and also conduct interviews with healthcare providers (see work streams (WS) 1, 2 and 3 further). Beyond the hospitals, the activity will take place in the rest of the SA, visiting AB retailers, households and communities. At the start of activities in all SAs, community inception workshops will be held to

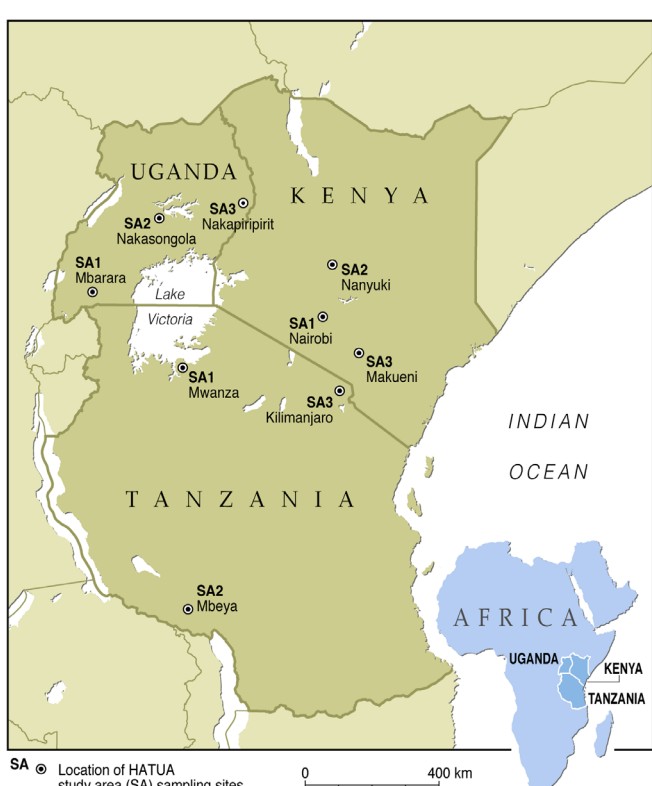

**Figure 2** Location of HATUA SAs. HATUA, Holistic Approach to Unravel Antibacterial Resistance in East Africa; SA, study area.

introduce and communicate the goals of HATUA to relevant stakeholders (the local community, village health teams, doctors, hospital and lab workers, and local health authorities). Sampling will begin in April 2019 in some sites and will continue through 2020 (COVID-19 permitting). All data collection tools and operating protocols are standardised, allowing valid comparison across sites and countries. For collecting quantitative, social science and laboratory data and geospatial mapping, we will use EpiCollect 5 (https://five.epicollect.net),[29] a customisable mobile data gathering tool installed on tablets and mobile phones.

### Recruitment of sample of patients with UTI
At the heart of the HATUA study will be a linked data set of 1800 patients (600 per country, 200 per site with culture-confirmed UTI (table 1). Given the estimated UTI prevalence from the pilot study, this will mean recruiting three times that number, c. 5400 patients. Sample size calculations were challenging, given the number of possible ways of measuring this, and limited evidence of different prevalence rates for community and hospital-acquired resistant UTI infections of adults and children in this region. To estimate precision, under a binomial model, the numbers required to obtain a 95% CI for the prevalence of 0.5 with width no greater than 0.1 would be a little under 400 (384). That model relies on there being no underlying population or sampling structure and so will lead to an underestimate of the true required

**Table 1** Target sample sizes for HATUA data collection tools

| | Per study site | | | Per country total | Study total |
| --- | --- | --- | --- | --- | --- |
| | Phase 1 | Phase 2 | Total | | |
| Patient questionnaire | ~300, to recruit 100 UTI+ | ~300, to recruit 100 UTI+ | ~600, to recruit 200 UTI+ | ~1800, to recruit 600 UTI+ | ~5400, to recruit 1800 UTI+ |
| Household questionnaire | 100 | 100 | 200 | 600 | 1800 |
| Patient IDIs | 5 | 5 | 10 | 30 | 90 |
| Healthcare worker IDIs | 1–5 | 0–5 to fill quota | 5 | 15 | 45 |
| Drug seller mystery client study visits | – | 50 | 50 | 150 | 450 |
| Drug seller IDIs | – | 10 | 10 | 30 | 90 |
| Focus group discussions | – | 4–8 | 4–8 | 12–24 | 36–72 |

HATUA, Holistic Approach to Unravel Antibacterial Resistance in East Africa; IDI, in-depth interview; UTI, urinary tract infection.

numbers in our complex study. Our larger study size of 600 per country will provide some robustness to our ability to estimate this parameter with the desired accuracy while allowing us to uncover some of the population structures that, if modelled correctly, will improve the precision in our estimate of prevalence. In level 2, 3, 4 and 5 hospitals in each SA, we will recruit adult and child outpatients (minimum of 90% of the total sample) that a doctor identifies as suffering with UTI-like symptoms (eg, burning/irritation during urination, dysuria and pyuria). In level 5 hospitals, we will also recruit inpatients (maximum of 10% of the total). For non-pregnant child patients aged under 18 years, data will be provided by an accompanying parent or guardian. Our sample is representative only of the population of clinic attendees rather than the general population and is likely to include a higher proportion of patients with treatment failures who are wealthier and patients living closer to clinics. However, clinic attendees are an important patient subset as these are the individuals specifically for whom clinicians must make patient management and treatment decisions.

A urine sample and, where possible, a faecal sample will be taken from all patients. From catheterised inpatients, urine catheter samples will be gathered, whereas

outpatients will be advised how to self-collect mid-stream urine samples. In addition, patients will have a questionnaire administered to collect retrospective data on their recent clinical history and related treatment seeking and AB usage (see figures 3 and 4 for the topics covered in the questionnaire). The questionnaire will capture individual-level sociodemographics (eg, age, gender, education, household and family circumstances), household socio-economic factors (eg, housing type, amenities and asset ownership used to derive multidimensional poverty indices), treatment-seeking behaviours, attitudes towards medication, ABs and AMR, and residential geographical information.

During initial recruitment, all patients will be asked to provide consent to being followed up if they test positive for a UTI. Eligible outpatients with culture-confirmed UTI will be recontacted for a follow-up to their homes (see WS4). For logistical reasons, only patients living within an approximate distance of 70 km from the level 5 hospital and 10 km from the level 4 and 3 hospitals will be eligible for follow-up. During follow-up a questionnaire will be administered to a competent adult member of the household to capture sanitation and hygiene, sociodemographics, economic and poverty dimensions, household

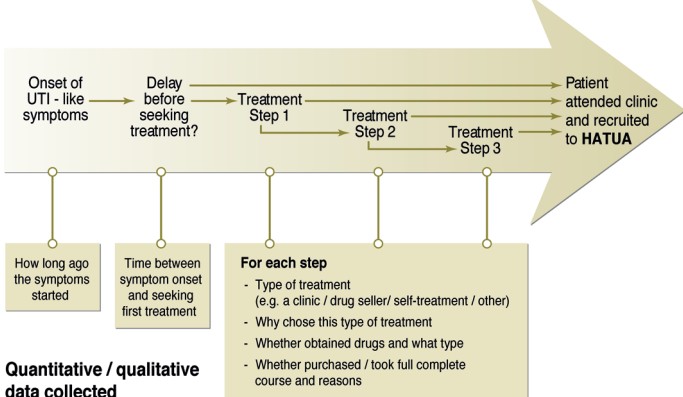

**Figure 3** Description of quantitative and qualitative data collected about the self-reported 'patient pathway' in the linked patient sample. HATUA, Holistic Approach to Unravel Antibacterial Resistance in East Africa; UTI, urinary tract infection.

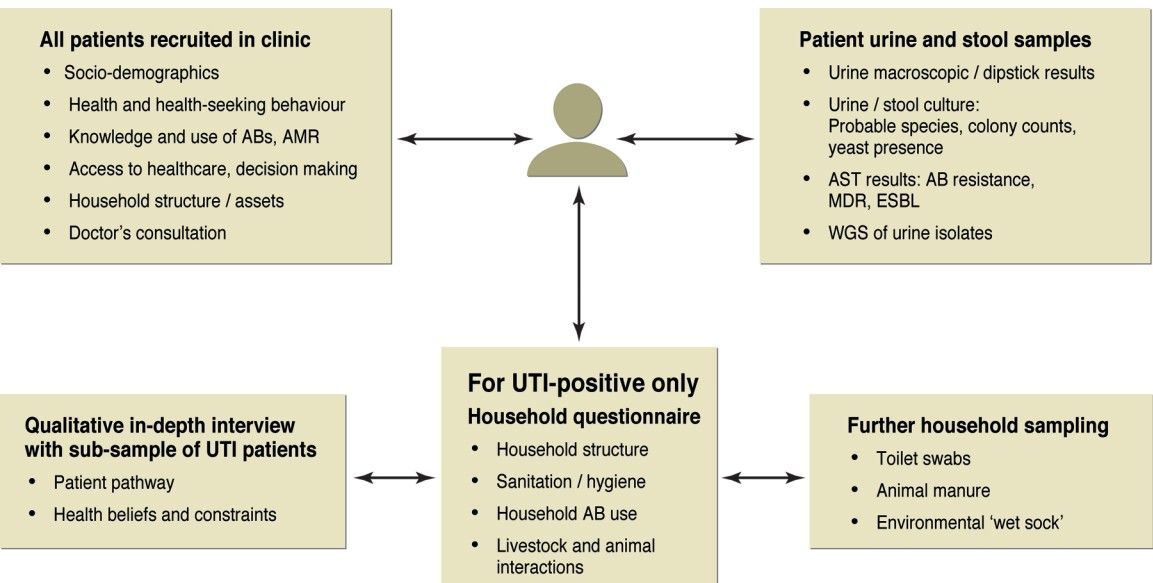

**Figure 4** Linked individual-level patient dataset collected in HATUA. AB, antibiotic; AMR, antimicrobial resistance; AST, antibiotic sensitivity testing; ESBL, extended spectrum beta-lactam resistance; HATUA, Holistic Approach to Unravel Antibacterial Resistance in East Africa; MDR, multidrug resistance; UTI, urinary tract infection; WGS, whole-genome sequence.

health-seeking behaviour, and livestock keeping practices. Environmental sampling of soil, animal faecal samples and other materials in the immediate proximity of the home will be conducted using a variety of approaches, including toilet swabs and boot sock/swab sampling of the soil in and around the homestead. To enrich the quantitative data, a subsample of patients with UTI will be purposively selected for qualitative IDIs based on their having drug-resistant UTI pathogens or reporting complex patient treatment pathways (10 per SA, 90 in total). The qualitative interviews will be conducted using a standardised topic guide across sites, which covers experience of illness, stigma, patient pathway narratives, experience of the doctor's consultation and subsequent steps, understanding of risk factors for UTI, AMR and AB stewardship.

The resulting qualitative and quantitative social science data and microbiological data will form an individual-level linked dataset. This will incorporate quantitative questionnaire data collected at the clinic and the home, qualitative data from IDIs, C&AST and whole-genome sequence (WGS) of pathogens from the patient, and C&AST from household samples (see figure 4). This can be related to multiscalar data on landscape (WS1).

### Further data collection
#### Geospatial mapping
Using EpiCollect on a GPS-enabled tablet, in all SAs, we will conduct geospatial mapping of observed AB providers in the local community (eg, from hospitals and clinics to retail pharmacies and informal drug sellers, to veterinary drug shops).

#### Mystery client study of drug sellers
A common means of reducing response bias from surveys is to use mystery client or simulated client studies to investigate 'real-life' dispensing practices of drugs sellers and pharmacies.[17 30] Using the sampling frame created by the aforementioned geospatial mapping exercise, we will randomly select outlets to participate in a simulated/mystery client study. Trained fieldworkers using predefined scenarios will request ABs and/or advice for the treatment of UTI-like symptoms. After the encounter, they will record data including whether and what kind of AB they were offered, the course and regimen they were sold, whether they were asked for a prescription, costs and advice given.

#### Qualitative IDIs with drug sellers
We will also select 10 drug sellers/ pharmacies per SA (see table 1) for qualitative IDIs to investigate their knowledge, motivations, attitudes and practices around AB provision and ABR. These will discuss the service they provide to the community, understanding and attitude to governance and AMR, and about the business and economic drivers of their work.

#### Qualitative IDIs with healthcare workers
In our recruitment hospitals/clinics, we will recruit trained clinical professionals using convenience sampling to investigate their knowledge and attitudes using qualitative IDIs (five per SA, 45 in total). The topic guide covers experience of UTI diagnosis and prescribing, knowledge, attitudes towards 'AB stewardship' and understandings of drivers of ABR, including economic and cultural issues.

## Community focus groups

In each SA, age-specific and gender-specific community FGDs will be conducted (among non-study participants), selecting from a range of socioeconomic status groups. The topic guide will cover experience of illness, pathways to care, experience of healthcare services, accessing ABs, perceptions of AMR and AB stewardship. In total, up to 24 FGDs per country will be conducted.

## Patient and public involvement

In addition to the community focus groups conducted during pilot work, and the community inception workshops (detailed above), at the end of SA activities, we will use Community Dialogues (CDs) that bring together community members, health workers and veterinarians to a face-to-face engagement discussing the findings, and stimulate community participation and full engagement, and also identify how grassroots health workers might be used most effectively to improve ABR stewardship.

## Research questions and data analysis plan

The main research questions and corresponding analyses are designed within five interlinked WSs.

### WS1: the therapy landscape

This WS will investigate how ABs are provided and used in our SAs in a number of interlinked ways. First, by analysing the geospatial data collected on AB providers, we will describe the spatial distribution and density of drug-dispensing outlets in both formal and informal healthcare settings. Second, we will analyse the data from the mystery client studies geospatially. Third, by combining quantitative statistical analysis of the mystery client study with systematic thematic coding of the qualitative interviews, we will investigate the AB provision practices and knowledge among different AB providers. By developing an understanding of differences in the AB provision landscape and the knowledge, motivations and practices of AB sellers, we will seek to determine what factors regulate individual patient pathways to AB use.

### WS2: pathogen

In this WS, we will confirm UTI, identify the pathogenic organisms present and determine the antimicrobial susceptibility. Urine samples from 1800 culture-confirmed patients with UTI will be analysed to investigate the burden of disease and resistance. Antibiotic sensitivity testing (AST) will be conducted on an agreed set of clinically relevant ABs, along with special phenotypes such as extended-spectrum beta-lactam resistance (ESBL) (see online supplemental appendix B). The WHONET data capture and reporting software will be adopted in all SA hub laboratories, providing automated analysis of various multidrug-resistant phenotypes.

Genomic DNA of the samples will be extracted and sequenced. The resulting WGS libraries will be used to characterise the isolates and define pathogen population structures. We will identify the local and regional spread of ABR determinants and describe their evolutionary dynamics and local reservoirs among HATUA bacterial populations. These data will be linked to patient and household sociodemographics to pinpoint possible drivers of resistance at patient, hospital and household levels. By comparing our collections with previously published genomes, high-risk clones and their potential origins will be determined and their spread mapped across space and time.

### WS3: patient

This WS will investigate social, structural and behavioural drivers of ABR by identifying the various patient pathways to treatment and how these intersect with the ABR process. We will summarise quantitative pathway data using longitudinal latent class analysis and/or sequence analysis, and relate this statistically to ABR profiles at individual and community levels. To identify how treatment pathways could become more clinically effective, we will combine quantitative and qualitative patient pathway data with IDIs from drug sellers and doctors to investigate sources of prescribed and non-prescribed ABs and practices, prevalence and determinants of self-medication, and how these are constituted.

### WS4: community

In this WS, we will investigate social and community attitudes to treatment seeking, AB use and ABR, and how community household level factors, including hygiene practices, health-related behaviour, livestock keeping practices and the household microbiological landscape, influence ABR and broader-risk burdens. Questions in the household questionnaire addressing AB use with animals and animal products will be statistically related to ABR burden in urine, faecal and environmental samples. We will also analyse relevant data gathered during patient IDIs and community focus groups, transcribed and translated by local translators and fieldworkers, using systematic thematic coding in NVivo to give us information about experiences of illness and localised rationales for treatment.

### WS5: interdisciplinary synthesis

In this WS, we will integrate and synthesise the data collected in WS1–4 to explore how population and individual level behaviours and processes interact to contribute to the risk of ABR. Using the patient linked dataset, we will generate hypotheses about direct and indirect drivers of ABR using Bayesian network analysis.[31] We will use Bayesian networks to identify latent factors in different data types and then connect them with each other and key outcome variables in a heterogeneous network across all data. The network structure will identify direct and indirect influences on ABR, and Bayesian networks' probabilistic inference will predict the probability of impact on ABR of change in different drivers. Multilevel regression will then be used to identify which of these direct and indirect drivers account for the most variance in outcome and provide numerical predictions

of modifications. Information about AB sensitivity of urinary pathogens will be blinded from fieldworkers conducting household visits and questionnaires and from microbiologists collecting environmental samples from households to ensure fidelity.

## ETHICAL CONSIDERATIONS
### Informed consent
Written informed consent will be obtained from all participants prior to any data or specimen collection, with the exception of drug sellers taking part in the mystery client study, for whom the process would invalidate the approach. Participants will provide consent for questionnaire-filling/interviewing/focus groups (including audio recording, which will be subsequently transcribed and translated by locally trained fieldworkers) and for sample collection and analysis, and shipment of samples to third-party labs for WGS. All patient samples will be non-invasive by urine and stool collection only. The principal language of recruitment and administration of the informed consent document (ICD) and the questionnaire will be the language used in hospitals (ie, Kiswahili for Kenya and Tanzania, and Luganda, Runyankole and Ngakarimojong for Uganda). Local translators will be used to draft the ICD, and ICD will be back -translated. The consent process will be administered by fieldworkers who understand the relevant languages and dialects.

### Privacy and confidentiality
A number of procedures will be used to protect the confidentiality of respondents and the information collected: (1) questionnaires/interviews will be conducted only in a private setting; (2) all interviewers will be trained in research ethics; (3) all data will be kept strictly confidential, and numerical IDs will be used in place of names on all of the data collection instruments. At each new step of data collection, participants will be informed of confidentiality procedures during the consent process. All patients in the linked part of the study will be anonymised and identified through an eight-figure identifier and barcode to link social science and laboratory data effectively. Any personal data will be stored on handwritten consent forms securely and separately from questionnaire, test results and patient IDIs. EpiCollect data are uploaded to a secure cloud server. The reason for recruitment to the household follow-up will not be disclosed or discussed with anyone except the patient/respondent themselves. Geospatial data identifying home locations and drug sellers will be edited to avoid disclosure. Participants in other IDI and FGDs, together with data from mystery client visits, will be anonymised, and recordings, translations and transcriptions will be stored and transferred securely.

### Ethics approvals
The study received ethical approval from the University of St Andrews, UK (number MD14548, 10/09/19); National Institute for Medical Research, Tanzania (number 2831, updated 26/07/19); CUHAS/BMC Research Ethics and Review Committee (number CREC /266/2018, updated on 02/2019); Mbeya Medical Research and Ethics Committee (number SZEC-2439/R.A/V.1/303030); Kilimanjaro Christian Medical College, Tanzania (number 2293, updated 14/08/19); Uganda National Council for Science and Technology (number HS2406, 18/06/18); Makerere University, Uganda (number 514, 25/04/18); and Kenya Medical Research Institute (04/06/19, Scientific and Ethics Review Committee (SERU) number KEMRI/SERU/CMR/P00112/3865 V.1.2). For Uganda, administrative letters of support were obtained from the district health officers to allow the research to be conducted in the respective hospitals and health centres.

## RESULT DISSEMINATION AND IMPACT
HATUA's data will be available via an interactive website that will collate and present ABR data from across the EAC overlaid with socioeconomic, microbiological and genome data. Data will be visualised and shared in Microreact (https://microreact.org).[32] The consortium is partnered with the East African Health Research Commission, a statutory organ of the East African Community, which will spearhead the integration of HATUA outputs into policy at the regional level. In the near term, this will allow doctors to access information they can use to improve diagnosis and prescription patterns based on resistance profiles prevailing locally. In the longer term, HATUA will lay a strong foundation for a regional surveillance initiative and will provide a vital resource for regional ABR policy formulation. The results have great potential to inform policy, improve clinical practice and build capacity for pathogen surveillance in the region. The novel linked microbiological, genomic and social science data will provide new insights into social drivers of ABR.

**Author affiliations**
[1]School of Biomedical Sciences, Makerere University, Kampala, Uganda
[2]Centre for Microbiology Research, Kenya Medical Research Institute (KEMRI), Nairobi, Kenya
[3]Department of Microbiology and Immunology, Catholic University of Health and Allied Sciences, Mwanza, Tanzania
[4]College of Humanities and Social Science, Makerere University, Kampala, Uganda
[5]Geography and Sustainable Development, University of St Andrews, St Andrews, UK
[6]School of Public Health, Catholic University of Health and Allied Sciences, Mwanza, Tanzania
[7]School of Medicine, University of St Andrews, St Andrews, UK
[8]Kilimanjaro Clinical Research Institute, Kilimanjaro Christian Medical Centre and Kilimanjaro Christian Medical University College, Moshi, Tanzania
[9]School of Biology, University of St Andrews, St Andrews, UK
[10]School of Mathematics and Statistics, University of St Andrews, St Andrews, UK
[11]Department of Medicine, Brigham and Women's Hospital, Boston, Massachusetts, USA
[12]Clinical Research Department, London School of Hygiene & Tropical Medicine, London, UK
[13]Immunomodulation and Vaccines Programme, Medical Research Council/Uganda Virus Research Institute and London School of Hygiene & Tropical Medicine Uganda Research Institute, Kampala, Uganda

[14]Centre for Genomic Pathogen Surveillance, Wellcome Genome Campus, Cambridge, UK
[15]Big Data Institute, Li Ka Shing Centre for Health Information and Discovery, Nuffield Department of Medicine, University of Oxford, Oxford, UK
[16]East African Health Research Commission, Bujumbura, Burundi

**Acknowledgements** We thank Graeme Sandeman for assistance with drawing the figures.

**Collaborators** The Holistic Approach to Unravelling Antibacterial Resistance in East Africa Consortium, which includes all the aforementioned authors, plus: Catherine Kansiime (Makerere University, Uganda), Martha F Mushi (Catholic University of Health and Allied Sciences, Tanzania), Arun Gonzales Decano (University of St. Andrews, UK), Dominique L Green (University of St Andrews, UK), John Mwaniki (Kenya Medical Research Institute, Kenya), Nyanda E Ntinginya (NIMR - Mbeya Medical Research Centre, Tanzania), Joel Bazira (Mbarara University of Science and Technology, Uganda).

**Contributors** BBA contributed to the conceptualisation of the project, helped designed the tools, and led the pilot data collection and main data collection in Uganda. JK contributed to the conceptualisation of the project, helped design the tools, and led the pilot data collection and main data collection in Kenya. SEM contributed to the conceptualisation of the project, helped designed the tools, and led the pilot data collection and main data collection in Tanzania. SN contributed to the conceptualisation of the project, helped design the social science tools, led the pilot data collection and coordinated social science data collection in Uganda, and led WS4. KK contributed to the conceptualisation of the project, helped designed the social science tools, contributed to the analysis plan, led WS5 and wrote the first draft of this protocol. MK contributed to the conceptualisation of the project, helped design the social science tools and contributed to the analysis plan. JRM contributed to the conceptualisation of the project, helped design the social science tools, and coordinated social science data collection in Tanzania. DJS contributed to the conceptualisation of the project, helped design the tools and led WS3. BTM contributed to the conceptualisation of the project and helped coordinate data collection in Kilimanjaro, Tanzania. VAS contributed to the conceptualisation of the project, wrote parts of the data analysis plan and supervised the analyses. SHG contributed to the conceptualisation of the project, led WS2 and oversaw microbiological data quality. AGL advised on the statistical elements for the project and helped draft this protocol. AS coordinated data collection and work stream activities and helped write this draft protocol. JS contributed to the conceptualisation of the project and provided oversight to WS2. AE contributed to the conceptualisation of the project, and helped draft this protocol. DMA contributed to the conceptualisation of the project, and provided the genomic analysis for WS2. GEK contributed to the conceptualisation of the project and facilitated policy dissemination in EAC. WS contributed to the conceptualisation of the project, helped design the tools and led WS1. MTGH led the conceptualisation of the project, helped designed protocols and data collection tools, and was the guarantor of the project. All authors revised the draft and revised versions of the paper.

**Funding** The Holistic Approach to Unravel Antibacterial Resistance in East Africa is a 3-year Global Context Consortia Award (MR/S004785/1) funded by the National Institute for Health Research, Medical Research Council and the Department of Health and Social Care. The award is also part of the EDCTP2 programme supported by the European Union. The funders had no role in study design, data collection and analysis, decision to publish or preparation of the manuscript. This work is supported in part by the Makerere University-Uganda Virus Research Institute Centre of Excellence for Infection and Immunity Research and Training (MUII). MUII is supported through the DELTAS Africa Initiative (grant number 107743). The DELTAS Africa Initiative is an independent funding scheme of the African Academy of Sciences and Alliance for Accelerating Excellence in Science in Africa, and is supported by the New Partnership for Africa's Development Planning and Coordinating Agency with funding from the Wellcome Trust (grant number 107743) and the UK Government. This paper was funded in part by a grant from the National Institutes of Health (grant number U01CA207167).

**Map disclaimer** The depiction of boundaries on the map(s) in this article does not imply the expression of any opinion whatsoever on the part of BMJ (or any member of its group) concerning the legal status of any country, territory, jurisdiction or area or of its authorities. The map(s) are provided without any warranty of any kind, either express or implied.

**Competing interests** None declared.

**Patient consent for publication** Not required.

**Provenance and peer review** Not commissioned; externally peer reviewed.

**ORCID iDs**
Katherine Keenan http://orcid.org/0000-0002-9670-1607
Alison Elliott http://orcid.org/0000-0003-2818-9549

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
