## [Reviewer comments · BMJ Open]

ARTICLE DETAILS

TITLE (PROVISIONAL)	Protocol for an interdisciplinary cross-sectional study investigating the social, biological and community-level drivers of antimicrobial resistance (AMR): Holistic Approach to Unravelling Antibiotic Resistance in East Africa (HATUA)
AUTHORS	Asiimwe, Benon; Kiiru, John; Mshana, Stephen E.; Neema, Stella; Keenan, Katherine; Kesby, Mike; Mwanga, Joseph R.; Sloan, Derek; mmbaga, blandina; Smith, V Anne; Gillespie, Stephen; Lynch, Andy G.; Sandeman, Alison; Stelling, John; Elliott, Alison; Aanensen, David; Kibiki, Gibson E.; Sabiiti, Wilber; Holden, M; Consortium, HATUA

VERSION 1 – REVIEW

REVIEWER	A/Prof Mitchell K Byrne University of Wollongong, Australia
REVIEW RETURNED	04-Aug-2020

GENERAL COMMENTS	This is an excellent project, linking biological, social sciences, and geographical data to answer a pressing problem: AMR. The theory section in the introduction was particularly sound. There are, however, a number of minor and moderate problems to resolve before publication. Minor 1. Page 4, line 50 - define acronym LMIC2. Page 5, line 6 - after colons (:) should follow semi-colons (;) in lists Page 6, line 16 - insert the word Fig 1 in parentheses for consistency and clarity 3. The authors need to be consistent in the use of ABR vs AMR (ABR for this study)4. Page 7, line 52 "UTI infections OF adults..." Medium 1. The Pilot Phase (p. 6-7) requires a data summary to evidence (a) feasibility and (b) how the pilot information was used to inform the main study, including sample size considerations2. There is an absence of information/detail on questionnaires to be used, environmental sampling techniques and analytic processes, and of the structure of in-depth interviews (with respect to the assessment of knowledge, attitudes, motivations and practices)3. Page 14, line 12 indicates that consent will be obtained for audio recordings but does not describe transcription methods4. Page 12 - aspects of the ethical considerations better placed in methods5. No discussion of whether aspects of the data will be blind to investigators before analysis to ensure fidelity
--

REVIEWER	Brooke Ramay Washington State University, USA Universidad del Valle de Guatemala, Guatemala
REVIEW RETURNED	10-Oct-2020

GENERAL COMMENTS	The protocol represents a very complete "holistic approach" to identifying drivers of AMR. The study objectives, design and methodologies are very clear. I look forward to reading about the results of this study.
--

REVIEWER	Dr Joseph O. Fadare Department of Pharmacology and Therapeutics College of Medicine Ekiti State University Ado-Ekiti Nigeria
REVIEW RETURNED	24-Dec-2020

GENERAL COMMENTS	The study which addresses AMR using several approaches is well conceptualized and will likely
---

VERSION 1 – AUTHOR RESPONSE

Response to Reviewers

Manuscript ID bmjopen-2020-041418

Title: "Protocol for an interdisciplinary cross-sectional study investigating the social, biological and community-level drivers of antimicrobial resistance (AMR): Holistic Approach to Unravelling Antibiotic Resistance in East Africa (HATUA)."

Reviewer: 1

This is an excellent project, linking biological, social sciences, and geographical data to answer a pressing problem: AMR. The theory section in the introduction was particularly sound. There are, however, a number of minor and moderate problems to resolve before publication.

Minor

1. Page 4, line 50 - define acronym LMIC

Response: this has been changed to low-and middle-income countries

2. Page 5, line 6 - after colons (:) should follow semi-colons (;) in lists

Response: this has been changed.

3. Page 6, line 16 - insert the word Fig 1 in parentheses for consistency and clarity

Response: this has been changed.

4. The authors need to be consistent in the use of ABR vs AMR (ABR for this study)

Response: This has been amended so that the term AMR is used in the introduction, when speaking about the field in general, but when we refer to this study, we now use ABR consistently throughout.

5. Page 7, line 52 "UTI infections OF adults..."

Response: this has been changed.

Medium

1. The Pilot Phase (p. 6-7) requires a data summary to evidence (a) feasibility and (b) how the pilot information was used to inform the main study, including sample size considerations.

We have elaborated the section on p6-7 which summarises the data collected, and we added a table in the appendix (Table A1) characterising the patient quantitative sample. The full details of the microbiological and genomic analysis from the pilot data is reported in another paper, cited in this section. This section now reads:

HATUA pilot activities in 2017-18 in Uganda and Kenya aimed to develop capacity and demonstrate the feasibility of the holistic study design by conducting a study of microbiological and genomic features of urinary pathogens collected from clinic patients, combined with quantitative socio-demographic data. Patients with UTI-like symptoms were recruited in public clinics and hospitals in Nairobi, Kenya and Isingiro District, Uganda, who provided bacterial samples (most commonly urine, but also stool). In Uganda, 129 of these patients, or their guardians, also completed questionnaires capturing their socio-demographic features, antibiotic knowledge and behavioural characteristics (see summary in Appendix Table A1). While the sample size for the questionnaire was not sufficient for detailed statistical analysis, the descriptive data indicated a higher proportion of women recruited than men (71% female), and the majority of working age. Over half of respondents had taken medication in the last six months (55%), and of those who did most obtained medications from clinics or health centres and nearly one fifth (18%) from drug shops.

Urine samples were analysed using culture and antibiotic sensitivity testing (C&AST) and a total of 150 bacterial isolates (n=91 from Kenya and n=59 from Uganda) were genomically characterised, which confirmed high prevalence of uropathogenic strains E. coli and K. pneumoniae, and revealed high levels of multi-drug resistance mainly disseminated via clonal and horizontal transfer (the full results are reported here [28]). This exploratory pilot study illustrated the feasibility of collecting linked microbiological, genomic and socio-economic data, and highlighted important operational issues to ensure accurate linkage and follow-up to the homestead.

We also conducted in-depth interviews with healthcare providers (e.g. nursing assistants), and four focus group discussions with community members (mainly crop farmers and pastoralists) in Isingiro district, Uganda to explore behaviours and attitudes to AB use, and identify possible drivers of ABR for hypothesis generation. These data were analysed using thematic content analysis, which highlighted potential drivers of ABR. Among community members, these included including distrust and misuse of ABs, failure to complete treatment courses, human use of veterinary drugs and combined consumption of ABs and traditional medicines, which informed the development of the research questions in the main study.

- 2. There is an absence of information/detail on questionnaires to be used, environmental sampling techniques and analytic processes, and of the structure of in-depth interviews (with respect to the assessment of knowledge, attitudes, motivations and practices)*

The details of the patient questionnaire topics are included in Figure 3, on page 9 we have now included a note about this - *“(see Figs. 3 and 4 for the topics covered in the questionnaire)”*- and added further detail in the text. On page 10, we included further detail about the methodology of collecting environmental samples, *“Environmental sampling of soil, animal faecal samples and other materials in the immediate proximity of the home will be conducted using a variety of approaches, including toilet swabs, and boot sock/swab sampling of the soil in and around the homestead.”*

On page 10, we have also added further detail about the topics covered in the patient in-depth interviews (IDIs): *“The qualitative interviews will be conducted using a standardised topic guide across sites, which covers experience of illness, stigma, patient pathway narratives, experience of the doctor’s consultation and subsequent steps, understanding of risk factors for UTI, AMR and AB stewardship.”*

On page 10, we added detail for the in-depth interviews with drug sellers: *“These will discuss the service they provide to the community, understanding and attitude to governance and AMR, and about the business and economic drivers of their work.”*

On page 10, we added detail for Qualitative IDIs with healthcare workers: *“The topic guide covers experience of UTI diagnosis and prescribing, knowledge, attitudes towards ‘antibiotic stewardship’, and understandings of drivers of ABR including economic, and cultural issues.”*

On page 11, we added detail for community focus groups, *“The topic guide will cover experience of illness, pathways to care, experience of healthcare services, accessing antibiotics, perceptions of AMR, and antibiotic stewardship.”*

3. *Page 14, line 12 indicates that consent will be obtained for audio recordings but does not describe transcription methods.*

On page 13, we added a note to the sentence to describe the process of transcription:

“Participants will consent for questionnaire-filling/interviewing/focus groups (including audio recording, which will be subsequently transcribed and translated by locally trained fieldworkers”.

4. *Page 12 - aspects of the ethical considerations better placed in methods*

The placement was guided by the BMJ manuscript format, but we have renamed the ‘discussion’ section ‘Ethics and Dissemination’. It is still situated towards the end of the manuscript as this represents the process of developing the protocol- we developed the design, tools and analytical plan before applying for ethical approval. However, we are happy to reorder these sections if the BMJ Open editors would prefer us to.

5. *No discussion of whether aspects of the data will be blind to investigators before analysis to ensure fidelity*

This is an important consideration, but due to the need for analysts to be involved in linkage, and having multiple potential outcomes, would be complex to blind all outcomes to the analysts. However, we ensure that the drug resistance status of the patients is blind to those collecting data at the homestead stage and conducting in-depth interviews with patients. We add a note in the manuscript to this effect *“Information about antibiotic sensitivity of urinary pathogens will be blinded from fieldworkers conducting household visits and questionnaires, and from microbiologists collecting environmental samples from households, to ensure fidelity”.*

Reviewer: 2

The protocol represents a very complete "holistic approach" to identifying drivers of AMR. The study objectives, design and methodologies are very clear. I look forward to reading about the results of this study.

Thank you for these comments; no changes made.

Reviewer: 3

The study which addresses AMR using several approaches is well conceptualized.

Thank you for these comments; no changes made.